# Critical Narrative Review of the Applications of Near-Infrared Spectroscopy Technology in Sports Science

**DOI:** 10.3390/s25216798

**Published:** 2025-11-06

**Authors:** Carlos Sendra-Pérez, Alberto Encarnación-Martínez, Jose I. Priego-Quesada

**Affiliations:** 1Research Group in Sports Biomechanics (GIBD), Department of Physical Education and Sports, Universitat de València, 46010 Valencia, Spain; alberto.encarnacion@uv.es (A.E.-M.); j.ignacio.priego@uv.es (J.I.P.-Q.); 2Department of Education and Specific Didactics, Jaume I University, 12071 Castellon, Spain; 3Biophysics and Medical Physics Group, Department of Physiology, Universitat de València, 46010 Valencia, Spain

**Keywords:** muscle oxygenation, cycling, exercise testing

## Abstract

Near-Infrared Spectroscopy (NIRS) is a noninvasive technology used to monitor muscle oxygenation in sports science. Since its introduction in 1977, NIRS has evolved into a valuable tool for assessing physiological responses during exercise and rehabilitation. The history of NIRS dates back to early hemoglobin studies in the 19th century, with significant advancements in pulse oximetry during World War II. By the late 1980s, NIRS had become widely used in sports science, allowing researchers to evaluate muscle perfusion and metabolic thresholds in various activities. NIRS applications in sports include determining exercise thresholds, monitoring muscle oxygenation during training, assessing asymmetries between limbs, and evaluating mitochondrial capacity. Studies have explored its use in both team and endurance sports, highlighting its role in optimizing performance and preventing injuries. Beyond sports, NIRS technology is expanding into clinical fields, aiding in rehabilitation and patient monitoring. This critical review has identified several key areas for future research, including the need to clarify methodological influences, strategies to minimize the impact of adipose tissue on NIRS measurements, the importance of conducting longitudinal studies, increased research on sex-specific effects, and a greater emphasis on field-based studies. With continued advancements, NIRS is expected to further enhance our understanding of muscle physiology and human performance, making it a crucial tool in athletic performance assessment and clinical practice.

## 1. Introduction

Near-Infrared Spectroscopy (NIRS) is a noninvasive technology that emits light photons capable of penetrating tissues and organs in situ to monitor cellular events [1]. The light detected by the NIRS device results from the absorption characteristics of heme-containing compounds (i.e., oxyhemoglobin ([O_2_Hb]) and deoxyhemoglobin ([HHb]). However, since NIRS signals cannot distinguish between chromophores, measurements typically reflect the combined contributions of hemoglobin, myoglobin and cytochrome C oxidase, which accounts for approximately 2–5% of the signal [2,3,4]. The use of NIRS has proven valuable in both medical and sports science applications [5,6].

NIRS devices employed in sports science enable the extraction, either directly or indirectly, of various flux and oxygenation variables, which have been utilized in sports and research settings to monitor changes in muscle oxygenation status [2,3,5]. [O_2_Hb] represents oxygenated hemoglobin (i.e., oxy[Hb + Mb]), while [HHb] represents deoxygenated hemoglobin (i.e., deoxy[Hb + Mb]). Changes in total hemoglobin (THb) within the muscle are calculated as the sum of [O_2_Hb] and [HHb]. Additionally, muscle oxygen saturation (SmO_2_) expresses the proportion of [O_2_Hb] relative to THb on a 0–100% scale, providing a normalized indicator of the balance between oxygen delivery and consumption within the muscle [7]. However, muscle oxygenation refers to the physiological state of oxygen supply and utilization within muscle tissue; SmO_2_ represents a specific quantitative measure of that process derived from NIRS technology.

Although all these variables assessed during exercise training or testing could provide a better understanding of changes in local perfusion [3,4], discrepancies between blood hemoglobin concentrations and the THb measured by NIRS devices—along with the greater reliability of SmO_2_ have led researchers to recommend SmO_2_ as the most consistent and informative metric [8,9,10]. Nevertheless, the accuracy and reliability of NIRS-derived variables can be influenced by several technical and physiological factors. In particular, subcutaneous adipose tissue thickness affects signal quality by attenuating light transmission through the tissue, while the source–detector separation determines the effective penetration depth of the NIRS light.

This narrative and critical review aims to explore the applications of NIRS technology in sports science, highlighting its main strengths and limitations. Compared with other portable physiological tools (e.g., heart rate devices), NIRS technology provides direct, localized insights into muscle oxygen dynamics. Its combination of being lightweight, wireless, waterproof, and non-invasive makes it one of the few wearable technologies capable of capturing real-time muscle metabolism in both laboratory and field settings. Before addressing these aspects, a brief overview of the historical development of NIRS for muscle oxygenation measurement is presented. In 1876, Karl von Vierordt visually observed spectral changes in hemoglobin in human fingers when circulation was interrupted. Half a century later, in 1932, Drabkin and Austin constructed devices to perform in vitro spectrophotometry using visible light (400 to 650 nm) [11]. Later, during World War II, Glen Millikan developed the first portable oximeter using red light and NIRS to warn military pilots of dangerous hypoxia [12]. Earl Wood and J. E. Geraci modified the Millikan earpiece by incorporating Squire’s pneumatic cuff. Later, Takuo Aoyagi tested various wavelengths (630–900 nm) and methods to obtain a higher sensitivity to oxygen saturation for pulse oximetry [12]. A few years later, NIRS technology was introduced in 1977 by Professor Frans F. Jöbsis to measure oxygenation of the exposed heart and brain without surgical intervention [13]. To date, this technology has only been used to investigate large organs (e.g., the brain), and Jöbsis investigated other tissues, such as skeletal muscles, using NIRS technology [11]. Since the late 1980s, NIRS technology has been used in sports science research in different modalities, from team sports to individual endurance sports or even contact sports such as boxing [3,14]. However, the greatest evolution has occurred in the last 10 years with the emergence of portable, lightweight, compact, wireless, economically competitive systems with good repeatability [7,15] that allow for their improved application in sports science, both in laboratory and field settings.

In this sense, it was carried out an electronical search in a database (i.e., PubMed) using the following terms per title and abstract: “Near Infrared Spectroscopy” or “NIRS” and “muscle oxygenation” or “oxygenation”, and “exercise” or “sport” or “physical activity”. The Boolean operators “OR” and “AND” were used to combine within and between the search terms of the subject areas. Figure 1 shows the results for the online database. Since the 1970s, interest in NIRS technology has increased in the field of sports science, and the number of published studies in recent years has exponentially increased owing to the emergence of portable devices that are much cheaper than the initial NIRS sensors [5,13]. 

## 2. Applications of NIRS Technology in Sports

NIRS technology has become an increasingly valuable tool in sports science, enabling the non-invasive assessment of muscle oxygenation dynamics under diverse physiological and environmental conditions. To better organize its uses, these applications can be grouped into three interrelated domains: (1) performance assessment and (2) physiological adaptations.

### 2.1. Performance Assessment

-Thresholds determination during graded exercise testing in several sports (e.g., cycling, running, and rowing) and muscles (e.g., vastus lateralis or gastrocnemius) [15,16,17]. A recent meta-analysis suggests that determining the second threshold is relatively reliable, whereas the first threshold is somewhat less consistent [15]. Furthermore, there is no consistent evidence suggesting that local thresholds occur at different times than systemic thresholds [18]. Therefore, although NIRS can be considered an interesting non-invasive tool to determine the second metabolic threshold, there is a lack of evidence about its usefulness for determining specific thresholds for different muscles.-Physiological load assessment through muscle oxygenation responses during strength training (i.e., squat exercise) [19]. Although this application is promising, it remains unclear how factors like fatigue or muscle damage influence SmO_2_ responses. Evidence indicates that exercise-induced muscle damage can alter muscle oxygenation kinetics at rest and during exercise [20]. Likewise, acute high-intensity efforts can modify SmO_2_ dynamics in a task-specific manner and are associated with performance loss [21]. Future studies should consider whether specific protocols or exercises should be standardized to improve the reproducibility of internal load assessments.-Asymmetry limb assessment of muscle oxygenation responses in team [22] and endurance sports [23,24]. In cyclic sports like cycling, muscle oxygen saturation tends to be symmetrical across limbs [24]. However, individual differences have been observed, underscoring the potential utility of individualized assessments [24]. Future research should determine whether such asymmetries have clinical implications or reflect technical inefficiencies.-Critical Power, the percentage of SmO_2_ and its derivatives can be used to differentiate sustainable from unsustainable exercise intensities, describe and predict the depletion and repletion of the finite work capacity above critical power, and accurately estimate time to exhaustion [25,26].

### 2.2. Physiological Adaptations

-Monitoring improvements in muscle oxygenation during rehabilitation or training programs. For example, NIRS has been applied to evaluate muscle oxygenation in patients with stroke or chronic obstructive pulmonary disease during rehabilitation, showing potential for clinical application in monitoring functional recovery [27].-Evaluation of muscle oxygenation responses during intervention (e.g., the use of a tracksuit jacket with heating elements or supplementary inorganic nitrate) [28,29]. As with other physiological markers, NIRS provides a non-invasive means to assess differences in muscle metabolism under various experimental conditions.-Evaluation of muscle oxygenation changes during sport-specific tasks and physiological stressors (e.g., repeated sprint training or hypoxia training) [30,31]. Related to this topic, some researchers speculate about the possible detection of the slow component of VO_2_ using NIRS [32].-Determination of mitochondrial capacity through the assessment of muscle oxidative function (the maximum rate at which the muscle can utilize O_2_ to meet the energy demand of exercise) [33]. A widely used and reproducible method for this purpose is the 10 min occlusion test [33,34]. However, the literature reveals high variability in key methodological aspects such as occlusion pressure and duration [2]. Standardized protocols or expert consensus statements are needed to guide future applications.

NIRS technology is used in several applications, including sports. The last systematic review included 3435 athletes from more than 38 sports, from 27 studies focused on team sports (e.g., football, basketball, or rugby) to 55 studies that only employed the technology in cycling [5]. The following sections discuss the use of these technologies in team and endurance sports.

In addition to all the applications discussed, it is interesting to know which muscles are primarily being evaluated in the different studies, as this can be a good reference for starters in the use of NIRS. In a recent updated systematic review in 2024, authors showed how NIRS technology was employed in several muscles, see Table 1:

As shown in the literature, the vastus lateralis is the most commonly measured muscle using NIRS, primarily due to its superficial location and typically low subcutaneous adipose tissue thickness [5]. However, an increasing number of studies are now including alternative or multiple muscle sites, which opens up interesting avenues for exploring how muscle oxygenation profiles vary depending on the specific functional role of each muscle during a given exercise task [18,35].

The diversity of applications highlights the versatility of NIRS, yet also underscores the need for a unified framework to interpret muscle oxygenation data across different sports and contexts. The following sections explore how this technology has been applied in team and endurance sports, emphasizing both its potential and current methodological gaps.

## 3. NIRS Technology and Team Sports

We selected three representative studies [14,36,37] to illustrate the use of NIRS in team sports and the limitations associated with this technology.

In 2008, one of the first articles that used NIRS technology in a team sport was published by Kounalakis et al. (2008) [37], who examined anaerobic power and muscle oxygenation in elite male handball players. The study employed a 30 s Wingate test using arm cranking to assess upper-body anaerobic performance, comparing 21 top-level handball players with 9 physically active controls. The results demonstrated that handball players had significantly higher peak and mean power output, suggesting sport-specific adaptations in upper-body anaerobic capacity. Interestingly, the fatigue index was similar across groups, indicating that handball players maintained power output without experiencing greater fatigue. Using NIRS, the authors found that although both groups showed similar levels of muscle oxygen desaturation during exercise, the handball players exhibited significantly faster and more complete reoxygenation during recovery. For example, SmO_2_ returned to baseline in approximately 29.5 s in the handball players, whereas it failed to recover within 2 min in controls. Additionally, oxygenated and total hemoglobin levels were higher in handball players during recovery, suggesting enhanced muscle perfusion or oxygen delivery. Critically, the study provides compelling evidence of muscular and vascular adaptations in elite handball athletes, likely resulting from repeated high-intensity upper-body actions such as throwing, blocking, and pushing [37].

Following the early applications of NIRS in team sports, the technology was used in rugby and field hockey to assess muscle oxygenation during high-intensity exercise [14,36]. In the 2013 observational study by Jones et al. (2015) [36], six university-level rugby players performed a repeated sprint shuttle test designed to simulate the physiological demands of rugby sevens. The results showed consistent patterns of deoxygenation in the gastrocnemius during sprint bouts (maximum decrease of 24 ± 5%), followed by partial reoxygenation during rest periods, although with high interindividual variability. Some players demonstrated efficient recovery kinetics, while others exhibited impaired reoxygenation, failing to return to baseline even after extended rest. These findings underscore the potential of NIRS to detect acute hemodynamic responses during sport-specific activities but also highlight the challenges posed by individual variability, making it difficult to interpret results at the individual level without further validation.

Subsequently, a more robust intervention study by Jones et al. (2015) [36] applied NIRS to monitor changes in muscle oxygenation following a six-week sprint interval training program in elite female field hockey players. Although conducted in a different team sport, the findings are highly relevant to intermittent sports like rugby. Athletes who underwent sprint interval training showed significant improvements in speed performance and demonstrated enhanced muscle oxygen extraction, reflected in increased deoxy [Hb + Mb] and reduced SmO_2_. These changes occurred without significant alterations in total hemoglobin, suggesting improved local oxygen utilization without increased perfusion.

Critically, these studies demonstrate the sensitivity of NIRS to detect both acute physiological responses and chronic training adaptations in muscle oxygenation. However, they also reveal key limitations for its use in elite sport, such as large intersubject variability and not strong correlations between muscle oxygen saturation metrics and performance.

## 4. NIRS Technology and Endurance Sports

Since the 1990s, NIRS technology has been used in endurance sports to understand muscle metabolism. The endurance training increases the ability of skeletal muscles to utilize oxygen through several mechanisms [38]. Therefore, the study of oxygen transport and consumption is important for improving performance. It has been used in cycling and running during exercise testing in laboratory conditions to determine the metabolic threshold, oxygen extraction by a muscle, or to better understand oxygen transport during different exercise domains [16,17].

In the context of metabolic threshold determination, one relevant area of investigation is whether local thresholds differ from systemic ones (e.g., ventilatory or lactate thresholds). Boone et al. (2016) [3] proposed that rather than a direct correspondence, a cascade of events may occur between local thresholds—such as those derived from NIRS—and whole-body thresholds. However, this hypothesis remains inconclusive, and further research is needed. For instance, Caen et al. (2022) [39] found that local thresholds did not consistently align with systemic thresholds. Conversely, a more recent study reported that muscle oxygenation thresholds did not differ across muscles with distinct functional roles during cycling, nor did they differ from the lactate threshold [18]. To further investigate the “cascade of events” hypothesis, longitudinal studies are required—specifically, those involving interventions designed to shift thresholds—to determine whether local and systemic thresholds change in parallel or independently [40,41].

Despite these insights, NIRS technology has been primarily used to explore physiological events during exercise testing. Recently, its application in quantifying training load in endurance sports has gained attention [31,42], and NIRS has been used in field settings for several years. For example, muscle oxygenation in the vastus lateralis was monitored in 17 athletes during a time trial on an off-road trail course, and the main finding was that heart rate remained stable during the trail run and muscle oxygenation was lower in running uphill than downhill, inversely to the alternations found in oxygen uptake (VO_2_) [43]. This study demonstrated that muscle oxygenation, as a local physiological marker, may provide more sensitive insights into workload changes than systemic variables such as heart rate.

## 5. Limitations of NIRS

First, we outline several key limitations that, in our view, significantly hinder the application of NIRS technology in sports science.

A major issue is the lack of methodological studies ensuring the correct and standardized use of the technology. Many of these methodological aspects may seem minor, yet they are essential for data validity and reproducibility. For instance, the importance of skin preparation—such as shaving and cleaning the skin prior to sensor placement—remains unclear. Similarly, there is no consensus on the optimal site for sensor placement. While some studies follow surface electromyography protocols to determine the precise location [16,24], others base their decisions on the area of greatest muscle mass [44]. Another unresolved question is whether normalization procedures (e.g., expressing data as changes from baseline) are necessary to enhance the accuracy and reproducibility of NIRS measurements. These issues reflect critical methodological gaps that must be addressed to strengthen the scientific rigor and comparability of future studies. Therefore, a systematic comparison and harmonization of protocols across laboratories is crucial to enable meaningful inter-study comparisons and to consolidate evidence of NIRS in sports science.

Another key limitation is the influence of subcutaneous adipose tissue on NIRS signal quality. Several studies have shown that increased adipose tissue thickness can attenuate the NIRS signal, compromising measurement accuracy [2,45]. This significantly restricts the applicability of NIRS in sedentary or clinical populations, such as individuals with obesity. Addressing this limitation is essential to expand the use of NIRS for exercise monitoring in populations where fat mass is higher. One technological factor that may mitigate this issue is the optode separation distance—that is, the distance between the light emitter and receiver. A greater separation allows the signal to penetrate deeper tissues [2], potentially bypassing the attenuating effect of adipose tissue.

## 6. Future Studies

However, further research is needed to examine how different emitter-receiver separations influence NIRS signal quality as a function of adipose tissue thickness. Our narrative review proposes several lines of investigation that should be researched to determine whether signal normalization techniques can help reduce this confounding factor.

Finally, we propose several research directions that should be prioritized, following the resolution of the aforementioned methodological concerns:○Future studies should incorporate longitudinal designs to better assess training-induced changes.○More research is needed in female athlete populations, as sex-based differences in muscle oxygen saturation have been reported [46].○Further field-based studies should aim to link NIRS-derived metrics to actual performance outcomes during training, gameplay, and competition scenarios.

## 7. Conclusions

NIRS technology is a valuable tool for assessing muscle oxygenation in sports, aiding performance monitoring, training adaptations, and rehabilitation. Its noninvasive nature and real-time insights make it useful in both research and practice. Beyond sports, NIRS holds promise for clinical applications, with ongoing advancements likely to enhance its impact.

The significance of NIRS in training and rehabilitation lies in its ability to bridge physiological assessment and practical decision-making. NIRS-derived variables such as SmO_2_ can be integrated into training strategies to guide load management, optimize recovery intervals, and evaluate the effectiveness of conditioning programs. Similarly, in rehabilitation, NIRS offers a non-invasive approach to monitor muscle reoxygenation and detect improvements in tissue perfusion.

This critical review has identified several key areas that warrant further investigation, including the need to clarify methodological influences, strategies to minimize the impact of adipose tissue on NIRS measurements, the importance of conducting longitudinal studies, increased research on sex-specific effects, and a greater emphasis on field-based studies.

## Figures and Tables

**Figure 1 sensors-25-06798-f001:**
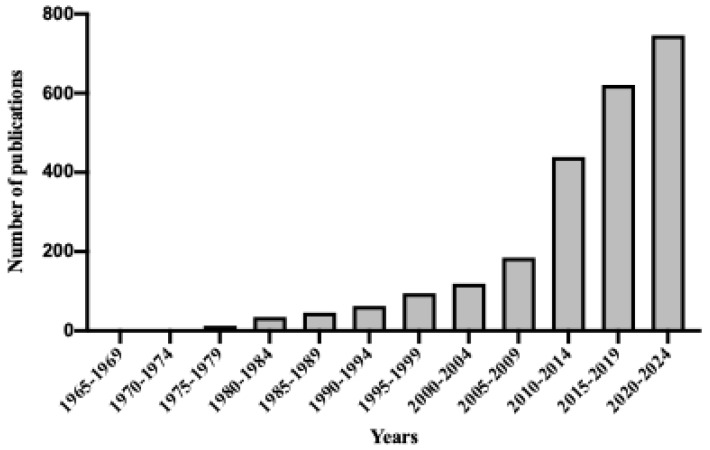
The number of studies published in PubMed using near-infrared spectroscopy (NIRS) to measure muscle oxygenation in sports science.

**Table 1 sensors-25-06798-t001:** Muscles evaluated employing the NIRS technology.

Muscle	Studies (N)
Vastus lateralis	138
Flexor fingers	20
Gastrocnemius medialis	20
Biceps brachii	12
Brachioradialis	12
Intercostal	9
Rectus femoris	7
Triceps brachii	6
Latissimus dorsi	5
Deltoid	3
Tibialis anterior	2
Biceps femoris	2
Trunk extensor	1

## Data Availability

No new data were created or analyzed in this study.

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
