# Peer review of "Critical Narrative Review of the Applications of Near-Infrared Spectroscopy Technology in Sports Science"

_sensors, 2025, doi:10.3390/s25216798_

Round 1

Reviewer 1 Report

Comments and Suggestions for Authors

I would like to commend the authors for their thorough and thoughtful paper: Critical narrative review on the applications of Near Infrared Spectroscopy technology in sports science (sensors-3926233). The manuscript successfully balances a critical perspective with recognition of the clear benefits and potential of mNIRS, making it both informative and engaging. While I found the paper to be well-structured and valuable to the field, I believe there are several important issues that should be addressed to strengthen the manuscript prior to publication.

A general comment to start; when discussing mNIRS in sport science it is important to put the technology, in its current form, in relation to other tools. What other physiological measurements are available in the current form mNIRS is. Current meaning, portable, lightweight, wireless, waterproof, watch compatible, non-invasive etc. Very few, almost exclusively HR. These characteristics is what it makes it interesting to sport science, if the signal is reliable and provides robust meaning.

  1. If the authors are going to make an introduction on the distinction or lack of, of chromophores between Hb and Mb, all components should be mentioned and literature provided.

Including Cytochrome C oxidase. Which is normally considered to be negligible but is 2-5% of the signal. Given how relevant Cytochrome C is for metabolism understanding and possibly separating this signal has future potential.

  1. In line 48 you have the term muscle oxygenation saturation(SmO2). This is incorrect terminology and grammar. It is muscle oxygen saturation (SmO2) or muscle oxygenation. Important is that these two things are often used interchangeably but are not the same thing. And given the content of this review, this should be made clear. What is SmO2 and what is muscle oxygenation? SmO2 has very specific components as it talks about saturation in the muscle of oxygen.  Saturation is has a biological meaning and therefore assumes something like a percentage to discuss the concept of fullness and emptiness. Muscle oxygenation does not need this for example, and is more of a general term.

Again, given the topic of the review, and the authors are rightfully critical of some of the ambiguous methodology used, this is the same for language and terminology around muscle NIRS. Specifically, the terminology around the concept of oxygenation when discussed as a percentage. If discussed as a percentage and not as arbitrary units, which is often used, there is factual component which in theory is testable, about how saturated the oxygen carrying/holding components are. This questions then the interchangeable use of the terms like TSI, TOI, SmO2, etc. This is further complicated by the technological difference and limitations between CW, FD, TD devices. Clearly, TSI SmO2 and TOI should not be used interchangeably.

The terminology perspective of course then also addresses the question of tHB concentrations. For example, CW devices should not propagate to provide hemoglobin concentration numbers, on AU of tHB. Which the vast majority of publications properly state.

  1. Line 99 states:

“Although this application is promising, it remains  unclear how factors like fatigue or muscle damage influence SmO₂ responses.” It is okay to ask for more studies, and more data. I would however say that numerous studies exist on this front and deserve to be citied. For example, but not limited to:

  • DOI: 1139/h08-048
  • DOI: 1111/j.1475-097X.2007.00777.x
  • DOI: 10.1007/s42978-021-00139-9

Therefore, I also think that while science is never conclusive, there are very clear consistent trends/expectations to mNIRS response to acute fatigue and muscle damage.

  1. I think that the authors are missing a very important application of mNIRS in athletics and that is its ability to assess time to task failure, or at least potential in a critical power or critical speed framework. Numerous studies have documented this, and given the well prescribed two critical power model to delineate between exercise intensity domains, it seems relevant. Example studies are but not limited to:
  • DOI: 10.1152/japplphysiol.00058.2021
  • DOI: 10.1152/japplphysiol.00706.2022
  • DOI: 10.1007/s00421-025-05825-y

Author Response

I would like to commend the authors for their thorough and thoughtful paper: Critical narrative review on the applications of Near Infrared Spectroscopy technology in sports science (sensors-3926233). The manuscript successfully balances a critical perspective with recognition of the clear benefits and potential of mNIRS, making it both informative and engaging. While I found the paper to be well-structured and valuable to the field, I believe there are several important issues that should be addressed to strengthen the manuscript prior to publication.

Reply: We thank the reviewer for the positive and encouraging feedback. We appreciate the recognition of our balanced and critical approach. In response, we have refined the manuscript to improve clarity.

A general comment to start; when discussing mNIRS in sport science it is important to put the technology, in its current form, in relation to other tools. What other physiological measurements are available in the current form mNIRS is. Current meaning, portable, lightweight, wireless, waterproof, watch compatible, non-invasive etc. Very few, almost exclusively HR. These characteristics is what it makes it interesting to sport science, if the signal is reliable and provides robust meaning.

Reply: We appreciate this insightful suggestion. We have now introduced a brief comparison of mNIRS with other portable physiological monitoring tolos (Lines 64-68).

“Compared with other portable physiological tools (e.g., heart rate devices) NIRS technology provides direct, localized insights into muscle oxygen dynamics. Its combination of being lightweight, wireless, waterproof, and non-invasive makes it one of the few wearable technologies capable of capturing real-time muscle metabolism in both laboratory and field settings.”

If the authors are going to make an introduction on the distinction or lack of, of chromophores between Hb and Mb, all components should be mentioned and literature provided. Including Cytochrome C oxidase. Which is normally considered to be negligible but is 2-5% of the signal. Given how relevant Cytochrome C is for metabolism understanding and possibly separating this signal has future potential.

Reply: We thank the reviewer for this valuable comment. We have now included Cytochrome c oxidase (Lines 41-42).

In line 48 you have the term muscle oxygenation saturation(SmO2). This is incorrect terminology and grammar. It is muscle oxygen saturation (SmO2) or muscle oxygenation. Important is that these two things are often used interchangeably but are not the same thing. And given the content of this review, this should be made clear. What is SmO2 and what is muscle oxygenation? SmO2 has very specific components as it talks about saturation in the muscle of oxygen.  Saturation is has a biological meaning and therefore assumes something like a percentage to discuss the concept of fullness and emptiness. Muscle oxygenation does not need this for example, and is more of a general term.

Reply: We thank the reviewer for this valuable comment. We have modified the term (Lines 48-49). In addition, we have added information about the differences between muscle oxygenation and SmO2 (Lines 48-53).

Again, given the topic of the review, and the authors are rightfully critical of some of the ambiguous methodology used, this is the same for language and terminology around muscle NIRS. Specifically, the terminology around the concept of oxygenation when discussed as a percentage. If discussed as a percentage and not as arbitrary units, which is often used, there is factual component which in theory is testable, about how saturated the oxygen carrying/holding components are. This questions then the interchangeable use of the terms like TSI, TOI, SmO2, etc. This is further complicated by the technological difference and limitations between CW, FD, TD devices. Clearly, TSI SmO2 and TOI should not be used interchangeably.

The terminology perspective of course then also addresses the question of tHB concentrations. For example, CW devices should not propagate to provide hemoglobin concentration numbers, on AU of tHB. Which the vast majority of publications properly state.

Reply: We appreciate the reviewer’s detailed remarks concerning the terminology and device-related differences in muscle NIRS measurements. We fully agree that parameters such as SmO₂, TOI, and TSI originate from different measurement principles and should not be used interchangeably. However, in the present review, we focused specifically on studies employing continuous-wave NIRS devices commonly used in sports science.

  1. Line 99 states:

“Although this application is promising, it remains  unclear how factors like fatigue or muscle damage influence SmO₂ responses.” It is okay to ask for more studies, and more data. I would however say that numerous studies exist on this front and deserve to be citied. For example, but not limited to:

  • DOI: 1139/h08-048
  • DOI: 1111/j.1475-097X.2007.00777.x
  • DOI: 10.1007/s42978-021-00139-9

Reply: We would to thank the reviewer for this comment. We have added that studies in our study (Lines 121-124)

“Physiological load assessment through muscle oxygenation responses during strength training (i.e., squat exercise) [24]. Although this application is promising, it remains unclear how factors like fatigue or muscle damage influence SmO₂ responses. Evidence indicates that exercise-induced muscle damage can alter muscle oxygenation kinetics at rest and during the exercise [25]. Likewise, acute high-intensity efforts can modify SmO₂ dynamics in a task-specific manner and are associated with performance loss [26]. Future studies should consider whether specific protocols or exercises should be standardized to improve the reproducibility of internal load assessments.”

Therefore, I also think that while science is never conclusive, there are very clear consistent trends/expectations to mNIRS response to acute fatigue and muscle damage.

  1. I think that the authors are missing a very important application of mNIRS in athletics and that is its ability to assess time to task failure, or at least potential in a critical power or critical speed framework. Numerous studies have documented this, and given the well prescribed two critical power model to delineate between exercise intensity domains, it seems relevant. Example studies are but not limited to:
  • DOI: 10.1152/japplphysiol.00058.2021
  • DOI: 10.1152/japplphysiol.00706.2022
  • DOI: 10.1007/s00421-025-05825-y

 Reply: Thank you for your insightful comment. The application of NIRS within the power critical and we agree that this is a highly relevant. The use of NIRS to estimate time to task failure has shown strong potential in delineating exercise intensity domains, particularly around the boundary between heavy and severe domains. We have added a point about it using those references (Lines 133-136).

Reviewer 2 Report

Comments and Suggestions for Authors

The paper proposes an overview of the use of near-infrared spectroscopy (NIRS) in sports science, describing its fundamental mechanisms, application scenarios, and limitations in assessing muscle oxygenation and athletic performance. The paper is well-structured, methodologically sound, and offers significant contributions to the field. However, the manuscript requires some revisions before a possible publication. Below are the reviewer's comments and suggestions aimed at enhancing the clarity, rigor, and overall quality of the manuscript.

1 The manuscript lacks a unified conceptual framework, as it does not present a clear analytical chain.

2 The authors should address the differences in conclusions among various studies and compare the methodological shortcomings.

3 The manuscript overlaps with existing reviews, as it extensively cites and reiterates the content of Perrey & Ferrari (2018, 2024) (Ref. [4] and [5]).

4 The authors should discuss how differences in device parameters affect the consistency of results, such as “subcutaneous fat thickness affects signal quality” (lines 245–247) and “source–detector separation affects penetration depth” (lines 251–253).

5 The manuscript lacks a clear explanation of the significance of NIRS in the training strategy or rehabilitation assessment.

6 The manuscript contains repetitive content. For example,
(1)both Section 2 “Applications” and Section 3 “Team Sports” include discussions on threshold determination, oxygenation monitoring, and asymmetry analysis.

(2) the descriptions in lines 81–87 and lines 143–147 present overlapping information.

7 The authors should integrate and analyze the referenced literature and clearly propose future research directions or hypothesis verification approaches in the conclusion section.

8 The authors should include additional tables, figures, or statistical summaries to provide data visualization and enhance the readability of the manuscript.

9 The authors should ensure consistency in the formatting of the references throughout the manuscript.

Author Response

The paper proposes an overview of the use of near-infrared spectroscopy (NIRS) in sports science, describing its fundamental mechanisms, application scenarios, and limitations in assessing muscle oxygenation and athletic performance. The paper is well-structured, methodologically sound, and offers significant contributions to the field. However, the manuscript requires some revisions before a possible publication. Below are the reviewer's comments and suggestions aimed at enhancing the clarity, rigor, and overall quality of the manuscript.

 Reply: We would to thank the reviewer for this comment. We are grateful for their comments and helped to improve significatively our paper.

1 The manuscript lacks a unified conceptual framework, as it does not present a clear analytical chain.

 Reply: Thanks for the comment. We have unified the conceptual framework in several subsections and transitions.

Applications of NIRS technology in sports (Lines 174-178):

“The diversity of applications highlights the versatility of NIRS, yet also underscores the need for a unified framework to interpret muscle oxygenation data across different sports and contexts. The following sections explore how this technology has been applied in team and endurance sports, emphasizing both its potential and current methodological gaps”

2 The authors should address the differences in conclusions among various studies and compare the methodological shortcomings.

 Reply: We have added a text in the manuscript, that address the differences in conclusion of the various studies (Lines 270-274)

“These issues reflect critical methodological gaps that must be addressed to strengthen the scientific rigor and comparability of future studies. Therefore, a systematic comparison and harmonization of protocols across laboratories is crucial to enable meaningful inter-study comparisons and to consolidate evidence of NIRS in sports science.”

3 The manuscript overlaps with existing reviews, as it extensively cites and reiterates the content of Perrey & Ferrari (2018, 2024) (Ref. [4] and [5]).

 Reply: Thanks! We have done some changes in the manuscript.

4 The authors should discuss how differences in device parameters affect the consistency of results, such as “subcutaneous fat thickness affects signal quality” (lines 245–247) and “source–detector separation affects penetration depth” (lines 251–253).

 Reply: We have commented in the introduction but, our narrative review seeks to convey a general concept of the state of NIRS technology in sports science (Lines 58-62)

“Nevertheless, the accuracy and reliability of NIRS-derived variables can be influenced by several technical and physiological factors. In particular, subcutaneous adipose tissue thickness affects signal quality by attenuating light transmission through the tissue, while the source–detector separation determines the effective penetration depth of the NIRS light.”

5 The manuscript lacks a clear explanation of the significance of NIRS in the training strategy or rehabilitation assessment.

 Reply: We thank the reviewer for this valuable observation. To clarify the practical relevance of NIRS, we have expanded the information about it in the conclusions (Lines 304-309)

6 The manuscript contains repetitive content. For example,

(1)both Section 2 “Applications” and Section 3 “Team Sports” include discussions on threshold determination, oxygenation monitoring, and asymmetry analysis.

(2) the descriptions in lines 81–87 and lines 143–147 present overlapping information.

 Reply: Thank you for your valuable comment. We agree that some sections contained overlapping information, particularly regarding threshold determination and oxygenation monitoring. To address this, we have revised the manuscript to improve the focus and avoid redundancy.

7 The authors should integrate and analyze the referenced literature and clearly propose future research directions or hypothesis verification approaches in the conclusion section.

Reply: Thank you for your valuable comment. We have added a new section about the future studies (Lines 284-297).

8 The authors should include additional tables, figures, or statistical summaries to provide data visualization and enhance the readability of the manuscript.

 Reply: We appreciate the reviewer’s suggestion. In response, we have added a conceptual figure and table (Figure 1 and Table 1).

9 The authors should ensure consistency in the formatting of the references throughout the manuscript.

 Reply: We have carefully reviewed and revised the reference list to ensure full consistency in formatting throughout the manuscript. All references now follow the same citation style regarding authorship, journal titles, volume and issue numbers, page ranges, and DOI formatting.

Round 2

Reviewer 1 Report

Comments and Suggestions for Authors

The additions improve the paper, and in my paper can be publihsed. 

Reviewer 2 Report

Comments and Suggestions for Authors

No more comments.